# Contribution of buoyancy fluxes to tropical Pacific sea level variability

Patrick Wagner[1], Markus Scheinert[1], and Claus W. Böning[1]

[1]GEOMAR Helmholtz Centre for Ocean Research Kiel, Kiel Germany

**Correspondence:** Patrick Wagner (pwagner@geomar.de)

**Abstract.** Regional anomalies of steric sea level are either due to redistribution of heat and freshwater anomalies or due to ocean–atmosphere buoyancy fluxes. Interannual to decadal variability in sea level across the tropical Pacific is mainly due to steric variations driven by wind stress anomalies. The importance of air–sea buoyancy fluxes is less clear. We use a global, eddy permitting ocean model and a series of sensitivity experiments with quasi–climatological momentum and buoyancy fluxes to identify the contribution of buoyancy fluxes for interannual to decadal sea level variability in the tropical Pacific. We find their contribution on interannual timescales to be strongest in the central tropical Pacific at around 10° latitude in both hemispheres and also relevant in the very east of the tropical domain. Buoyancy flux forced anomalies are correlated with variations driven by wind stress changes, but their effect on the prevailing anomalies and the importance of heat and fresh water fluxes vary locally. In the eastern tropical basin, interannual sea level variability is amplified by anomalous heat fluxes, while the importance of fresh water fluxes is small and neither has any impact on decadal timescales. In the western tropical Pacific, the variability on interannual and decadal timescales is dampened by both heat and freshwater fluxes. The mechanism involves westward propagating Rossby waves that are triggered during ENSO events by anomalous buoyancy fluxes in the central tropical Pacific and counteract the prevailing sea level anomalies once they reach the western part of the basin.

## 1 Introduction

Sea level is an integrated measure that contains information about the water column as well as about ocean–atmosphere interaction. Global mean sea level variability (GMSL) is usually separated into two processes: changes in the ocean's total mass due to redistribution of water between the ocean and the atmosphere, land or cryosphere, and changes of the density of the water column which are associated with changes of subsurface temperature and salinity distributions. Sea level variability is not spatially uniform (e.g. Stammer et al. 2013). Instead, it is influenced by ocean dynamics that act to redistribute ocean mass, heat and freshwater (e.g. Zanna et al. 2019). These redistributions leave the GMSL unaffected but can have strong impacts on regional scales rendering the spatial patterns of sea level variability highly non–uniform (e.g. Merrifield and Maltrud 2011). Ocean–atmosphere heat and freshwater fluxes can additionally modulate spatial patterns of sea level variability (e.g. Piecuch and Ponte 2012; Forget and Ponte 2015).

The tropical Pacific is a prominent example of an area where regional sea level variability substantially deviates from the GMSL. Between 1993 and roughly 2012 sea level rise in the western part of the basin was three times as strong as the global mean rate while the eastern basin saw no or even negative trends. These anomalies have been attributed to gradually increasing trade winds since the early 1990s (Merrifield and Maltrud, 2011). As such, they are part of a multidecadal pattern of variability that can be related to climate modes such as Pacific Decadal Oscillation (PDO) or El Niño Southern Oscillation (ENSO) (Merrifield et al., 2012).

It seems clear that the better part of sea level variability in the tropical Pacific is due to adiabatic processes and can be understood in terms of redistribution of heat driven by wind stress changes (Timmermann et al., 2010; Piecuch and Ponte, 2011; Merrifield, 2011; Merrifield and Maltrud, 2011; McGregor et al., 2012; Merrifield et al., 2012; Moon and Song, 2013; Moon et al., 2013; Qiu and Chen, 2012). However, exchange of heat and freshwater between the ocean and the atmosphere can also affect sea level variability, as they change the density of the surface waters. They are commonly referred to as "buoyancy fluxes". Only a few studies have addressed the importance of such diabatic processes for sea level in the region. Piecuch and Ponte (2011) assessed the contribution of surface buoyancy fluxes to steric sea level variability and identified a few regions where the contribution of local buoyancy fluxes is not negligible. One of them is the warm pool region in the western tropical Pacific. Other studies identified the central tropical Pacific as another region where local buoyancy fluxes contribute to interannual sea level variability (Piecuch and Ponte, 2012; Forget and Ponte, 2015; Meyssignac et al., 2017). Piecuch et al. (2019) argue that local, latent heat fluxes are in particular relevant on decadal timescales and contributed to the recent reversal of sea level trends in the tropical Pacific since 2012. All these studies used either local budget calculations or ocean model sensitivity simulations to estimate the contribution of buoyancy fluxes. Fukumori and Wang (2013) chose a different approach in a semi–Lagrangian model study and found buoyancy fluxes to be the dominant driver of sea level trends between 1993 and 2004 in the western Pacific warm pool.

A detailed assessment of the relative contribution of buoyancy flux anomalies to sea level variability in the region is important for projecting future sea level trends in this region, since adiabatic and diabatic forcing mechanism might evolve rather differently. However, in particular for the pre–altimetry era and on decadal timescales, we currently lack a detailed picture on how buoyancy fluxes affect sea level in the tropical Pacific and which mechanisms are involved. By means of eddy–permitting ocean model experiments that allow us to individually apply buoyancy and momentum flux forcing to the underlying ocean and by decomposing the steric sea level component into thermosteric and halosteric contributions, we assess the importance of buoyancy fluxes on interannual to decadal timescales over the last 6 decades, analyse their interplay with the wind stress driven variability and determine the importance of heat vs. fresh water fluxes.

## 2 Model experiments

We use a global ocean general circulation model (OGCM) configuration of the "Nucleus for European Modelling of the Ocean" (NEMO) code version 3.6 (Madec and NEMO-team, 2016). The model uses a global tri–polar ORCA grid at 1/4° horizontal resolution. The vertical grid consists of 46 z–levels with varying layer thickness from 6 m at the surface to 250 m in the deepest levels. Bottom topography is interpolated from 2–Minute Gridded Global Relief Data ETOPO2v2[1] and represented by partial steps (Barnier et al., 2006). The model is forced with the JRA55-do forcing (Tsujino et al., 2018) which builds on the JRA55 reanalysis product but is adjusted relative to observational datasets.

Laplacian and bilaplacian operators are used to parameterize horizontal diffusion of tracer and momentum, respectively. The ocean model is coupled to the Louvain–La-Neuve sea–ice Model version 2 (LIM2–VP, (Fichefet and Maqueda, 1997).

To avoid spurious drifts of global freshwater content, all models use a sea surface salinity restoring with Piston velocities of 137 mm per day, which corresponds to a relaxation timescale of one year for the upper 50 m, and a freshwater budget correction that sets changes in the global budget to zero at each model timestep.

In addition to a hindcast simulation (O025-HC) that uses realistic, interannual forcing and is integrated from 1958 to 2016, three sensitivity experiments were conducted to single out the effect of heat and freshwater fluxes (i.e. buoyancy fluxes) as well as intrinsic variability on sea level variability on interannual to decadal timescales. In these experiments the interannual variability of the turbulent ocean–atmosphere fluxes is suppressed, either for all fluxes (O025-RYF90), for momentum fluxes (O025-B90), or buoyancy fluxes (O025-W90).

This requires a method to eliminate interannual variability from the forcing, and we followed the approach of a "repeated year forcing" (Stewart et al., 2020) to construct quasi–climatological atmospheric fields. Specifically, a 12–month period, that is "neutral" with respect to a range of climate indices (hence the term quasi–climatological), is taken from the JRA55-do dataset and used repeatedly for the computation of turbulent fluxes. We followed the recommendation from Stewart et al. (2020) and used the period from May 1990 to April 1991 to force 59 cycles of each sensitivity experiment to match the length of the hindcast. The transition is moved from April to May, rather than December to January, to avoid periods of high variability at high latitudes, and therefore abrupt changes. The reader is referred to Stewart et al. (2020) for further information regarding the repeated year approach. All model output is stored with monthly resolution and has been used previously for studies of Indian Ocean heat content (Ummenhofer et al., 2020) and marine heatwaves (Ryan et al., 2021).

Three caveats need to be considered when analysing the sensitivity experiments: firstly, the computation of turbulent fluxes does not only depend on the atmospheric forcing but also the state of the underlying ocean. This introduces a source of interannual variability of buoyancy and momentum fluxes even where we aim to suppress it. Secondly, all experiments contain intrinsic variability, which is generated spontaneously by the ocean rather than by the atmospheric forcing. OGCMs with an

---

[1]http://www.ngdc.noaa.gov/mgg/global/relief/ETOPO2/ETOPO2v2-2006/ETOPO2v2g/

eddy–permitting resolution, as the ones used here, are able to capture this variability Penduff et al. (2011). Consequently, variability in O025-W90 for example, can not exclusively be attributed to momentum flux but might also be generated by intrinsic variability. We will use the climatological experiment O025-RYF90, where this will be the dominant source of variability, to quantify this. Thirdly, the approach assumes a linear superposition of variability forced by momentum and buoyancy fluxes. However, the OGCM used here does also include nonlinear responses of the ocean to atmospheric forcing, for example due to the non–linear nature of the equation of state, which may violate this assumption.

Interannual sea surface height[2] (SSH) variability from the model hindcast compares well with observations. Figure 1 shows interannual SSH variability as observed by satellite altimetry (gridded product is provided by the Copernicus Marine Service Center[3]) and simulated by the OGCM. Because the Boussinesq model used here does not capture the GMSL signal and the freshwater budget correction forces the GMSL anomaly to zero, the global mean signal has been removed from the altimetry data. The domain is characterized by the well–known zonal dipole with its eastern pole centred on the equator in the central to eastern Pacific. The western pole shows two maxima located around $10°$ S and $10°$ N in the western part of the basin. Observations show amplitudes up to 10 cm in the west and around 6 cm in the east (Fig. 1a). Overall, this structure is well captured by the model experiment, although the overall strength of the dipole is slightly underestimated with amplitudes of 8 cm and 5 cm in the western and eastern part respectively (Fig. 1b). SSH anomalies, averaged over three representative boxes in the northwestern ($0°$ N–$20°$ N, $130°$ E–$150°$ E), southwestern ($20°$ S–$5°$ S, $159°$ E–$179°$ E) and eastern ($7°$ S–$3°$ N, $210°$ E–$240°$ E) tropical Pacific (shown as red boxes 1–3 on the maps in Fig. 1), from observations and model simulation confirm the reduced amplitude of interannual variability by about 20%. However, the phase of the interannual variability is reproduced well, with correlation coefficients above 0.93 for all three boxes. Within a latitudinal band of about $40°$ around the equator, this result is not overly sensitive to the choice of the boxes (not shown). However, likely due to an insufficient spatial resolution, the model does not capture the mesoscale activity of the western boundary current regions properly and therefore underestimates SSH variability in this region even further, as can be seen by comparing Fig. 1 a) and b).

## 3  Results

Regional SSH variability is predominantly due to changes in the density of the water column (Fig. 2) with contributions from changes in the mass distribution negligible everywhere, except in shallow coastal regions (not shown). Fig. 2 shows the total SSH signal and its decomposition into steric, thermosteric and halosteric contributions for all three experiments. The steric signal in O025-HC is to a large extent due to changes in heat content (compare Fig. 2d and g). Compared to the total steric signal, the thermosteric variability shows slightly higher amplitudes in particular southwestern tropical Pacific. The same area is characterized by elevated values of halosteric variability (Fig. 2j), reflecting a compensating effect of thermosteric and halosteric

---

[2]We will refer to the sea level estimate of the OGCMs, which gives height above the geoid, as SSH

[3]GLOBAL OCEAN GRIDDED L4 SEA SURFACE HEIGHTS AND DERIVED VARIABLES REPROCESSED; https://resources.marine.copernicus.eu/?option=com_csw&view=details&product_id=SEALEVEL_GLO_PHY_L4_REP_OBSERVATIONS_008_047

changes in the region.

The momentum flux experiments yield a very similar result. Here, total SSH variability is mostly due to heat content changes with small contributions from halosteric changes that tend to dampen the thermosteric signal (compare Fig. 2b, e, h, k). Overall, these results concur with earlier accounts of the dominant role of thermosteric changes due to wind stress variability for SSH

variability in the tropical Pacific (e.g. Timmermann et al. 2010; Piecuch and Ponte 2011; Merrifield et al. 2012; Moon and Song 2013; Forget and Ponte 2015).

However, although interannual SSH variability driven by surface buoyancy fluxes is small compared to the momentum flux driven variability, it is not negligible. Its spatial pattern differs from the wind stress driven component (Fig. 2c). The buoyancy

flux contribution is most pronounced in the southern part of the study domain, with its maximum in the central Pacific around 10° S and is lowest on the equator. The buoyancy flux driven signal is again steric in nature but in contrast to the wind stress driven signal the halosteric contribution is comparable in magnitude to the thermosteric part (compare Fig. 2 f, i and l). Note that O025-B90 does also show strong variability in the boundary current region of the Kuroshio, north of 20° N. This is a region of strong intrinsic variability (Fig. 3), most likely due to the strong mesoscale activity. We will therefore neglect this region in

our analysis. It is also evident from Figure 3 that intrinsic variability is mostly negligible equatorward of approximately 20° latitude. Here, the ration between SSH variance from O025-RYF90 and O025-HC does not exceed 5%.

In order to assess the interplay between momentum and buoyancy fluxes, we analyse the change in variability between the hindcast experiment O025-HC and the momentum flux experiment O025-W90. This allows us to identify regions where mo-

140 mentum and buoyancy fluxes are correlated and to assess whether they tend to amplify or dampen one another. Figure 4 shows the change of interannual SSH variability in O025-W90 compared to O025-HC in terms of standard deviation (SD). Red/blue areas indicate regions where the variability increases/decreases when the interannual variability from the buoyancy flux forcing is removed, i.e. where buoyancy fluxes dampen/amplify wind stress driven SSH variability. The change in total SSH shows a pattern that is almost symmetrical about the equator (Fig. 4a). Negative values prevail around the equator and east of 225° E.

A horseshoe shaped minimum is located at the eastern boundary, where variability is reduced by up to 1 cm in the very east. Positive values are centred at 175° E on both sides of the equator at 10° latitude with values of up to 0.4 cm in the north and 0.6 cm in the south. The same pattern can be seen for steric SSH (Fig. 4b) and, to a slightly lesser extent, for thermosteric SSH (Fig. 4c). The change in halosteric SSH is similar in magnitude but shows a different pattern. Changes are mainly limited to western and southwestern tropical Pacific, where the halosteric SSH variability decreases in O025-W90 by up to 0.8 cm. Buoyancy

fluxes seem to have a small effect on halosteric SSH in the eastern to northeastern tropical Pacific, where SD in O025-W90 changes reach only 0.2 cm. Notably, regions where halosteric SSH variability is increased coincide with regions where total steric variability is damped. This indicates again the compensating effect of halosteric and thermosteric SSH changes. In short, buoyancy fluxes tend to dampen SSH variability in the eastern part of the domain and amplify it in the central and western part

of the region.

We take a closer look at two regions where buoyancy fluxes have a strong, but opposite effect and analyse spatial averages over the two boxes marked in Fig. 4. In the southwestern tropical Pacific, SSH changes driven by momentum and buoyancy fluxes are clearly out–of–phase on interannual to decadal timescales. This negative correlation (p=-0.6) corresponds to the damping effect of buoyancy fluxes identified before. Measured by the change in SD between O025-HC and O025-W90, buoy-
ancy fluxes dampen the variability by 11% on interannual and by 20% on decadal timescales. (Fig. 5a). The superposition of anomalies from O025-W90 and O025-B90 should reconstruct anomalies from O025-HC. We find a root–mean–square–error (RMSE) of 0.45 cm for this reconstruction. As discussed in section 2, possible causes are intrinsic variability (green lines in Fig. 5) and non–linear effects. Intrinsic variability is weak (SD of 0.11 cm), which suggest that non–linearities are the main cause.
Moving to the eastern box, we find steric SSH variability to show similar amplitudes of interannual and decadal variability with the corresponding signals being in phase (p=0.7). This is in line with the amplifying effect of buoyancy fluxes described above, and we find an increase of variability due to buoyancy fluxes by 16%. In contrast to the western box, buoyancy fluxes have no effect on decadal SSH variability in the eastern part of the basin (Fig. 5b). Here, the superposition of O025-B90 and O025-W90 to reconstruct O025-HC gives a much smaller RMSE of 0.2 cm. Unlike in the western box, intrinsic variability can
account for a large fraction of this (SD of 0.14 cm).
Additionally, Fig. 5 b) visualizes the expected strong correlation between SSH anomalies and ENSO–cycles.

Figure 2 suggests that both heat and freshwater fluxes drive SSH variability, in particular in the southwestern tropical Pacific. We investigate this further by decomposing SSH from O025-B90 into their thermosteric and halosteric parts again. Figure 6
shows this decomposition for SSH anomalies in the southwestern and eastern tropical Pacific. In the southwestern domain (Fig. 6a) steric SSH is influenced in equal parts by halosteric and thermosteric contributions. Correlation coefficients are 0.62 and 0.48 (both significant on a 99% confidence interval) respectively, and from the beginning of the simulation until approximately 1990 the variability of both components has a similar amplitude. However, thermosteric SSH variability decreases after 1990 by more than 20% (SD drops from 0.75 cm to 0.59 cm and a moving window calculation yields a linear decline of SD 0.1 cm
per 10 years that starts around 1985; Not shown) and steric SSH is subsequently primarily controlled by halosteric SSH (SD of 0.76 cm throughout the integration period). Steric SSH in the eastern equatorial Pacific is almost exclusively governed by thermosteric SSH with correlation coefficients of 0.89 (significant on a 99% confidence interval). While its variability is also smaller during the later part of the simulation, the decrease is not as drastic as in the western box (SD reduces from 71 cm to 63 cm). Halosteric SSH shows no significant correlations (on a 99% confidence interval) and only minor contributions, in
particular during El Niño events (e.g. the 1997/98 El Niño; SD of halosteric SSH increases from 0.20 cm to 0.25 cm between the two periods, which is due to this strong El Niño event).

The effect of buoyancy fluxes on SSH variability is uniform across the zonal extent of the tropical Pacific. This is in particular true south of the equator and explains the opposite effect of buoyancy fluxes (i.e. damping in the west and amplification in the east) on the wind stress driven zonal dipole structure. Figure 7 shows the temporal evolution of SSH along a zonal section at 10° S. The dipole structure in thermosteric SSH is clearly visible in O025-HC and O025-W90 (Fig. 7a, b, d, e) whereas the effect of halosteric SSH in the same experiments is limited to the central and western part of the region (Fig. 7c, f). Its compensating effect, related to the adiabatic advection of warm and saline surface waters, is clearly visible. Negative SSH anomalies dominate in the western part of the area during the negative PDO phase in the 1980s and 90s and are caused by thermosteric anomalies. Positive halosteric anomalies act to increase SSH. SSH anomalies in O025-B90 show a zonally uniform structure across the basin (Fig.g–j), i.e. their effect on the western and eastern poles of the wind stress driven, thermosteric dipole is exactly opposite. Maximum anomalies are located between 200° E and 220° E and are more pronounced in the western than in the eastern part of the area. The relative importance of buoyancy fluxes is highest for halosteric SSH, where amplitudes driven by momentum and buoyancy fluxes are comparable in magnitude (compare Fig 7f and i).

Our set of experiments specifically suggests that buoyancy–forced SSH variability in the tropical Pacific is relevant on interannual to decadal timescales (Fig. 2 and 5). Since the early 1990s, buoyancy flux driven SSH in the southwestern tropical Pacific is mostly controlled by halosteric effects, with the strongest signals emerging during El Niño or La Niña events (Fig. 6). In order to obtain insight into the mechanism by which halosteric SSH anomalies are triggered by buoyancy fluxes, how they spread zonally and how they interact with the wind driven SSH anomalies during ENSO events, we examine the particularly strong El Niño event in 1997/98.

## 4   Case Study: 1997/98 El Niño

SSH anomalies (defined as the deviation from the seasonal climatology) from O025-HC during the El Niño event show the characteristic zonal dipole (Fig. 8a). In contrast to that, O025-B90 shows positive SSH anomalies centred in the central basin at the beginning of 1998 (Fig. 8a) which propagate westward with phase speeds of about 17 cm s$^{-1}$, which is in accordance with phase speeds of Rossby waves derived from linear theory (Killworth et al., 1997). Once they reach the western part of the basin by mid–1998, these positive anomalies tend to reduce the prevailing negative anomalies (Compare fig. 8a and b). Anomalies in O025-B90 are mostly halosteric in nature (Fig. 8c). Areas of high precipitation start to propagate eastward at the beginning of 1997 and reach their easternmost position by the end of the same year where they trigger the SSH anomalies (Fig. 8d). There are only small precipitation anomalies east of 240° E, which explains the weak halosteric SSH anomalies in the eastern part of the basin described above. We note that the freshwater flux is controlled by precipitation at this latitude and evaporation is negligible (not shown). Thermosteric sea level anomalies are smaller than their halosteric counterpart (Fig. 8e) and heat flux anomalies do not show a coherent structure (Fig. 8f).

The co-occurrence in this example suggests that the halosteric SSH anomalies are at least partly driven by fresh water fluxes.

However, buoyancy fluxes in general could also change the flow field and thereby cause adiabatic temperature and salinity anomalies, that would also result in corresponding SSH anomalies.

A different picture emerges along the equator. The zonal SSH dipole in O025-HC (Fig. 9a) is related to positive anomalies in the eastern basin, which are amplified by buoyancy fluxes (Fig. 9b). The steric anomalies are dominated by their thermosteric component (Fig. 9b,e), triggered by positive heat flux anomalies that evolve throughout the zonal extent of the basin after mid–1997 and early 1998 (Fig. 9f). Precipitation anomalies between mid–1997 and mid–1998 (Fig. 9d) are even stronger at the equator than at $10°$ S but steric SSH anomalies in the central basin are weaker, and we do not find a pronounced zonal propagation of these anomalies. A possible reason is that sea level and thermocline depth expressions of Rossby waves are stronger off the equator, rendering signals of zonal propagation weaker on the equator. Another possible reason concerns the monthly resolution of the model output, which is insufficient to show the adjustment process associated with the eastward propagation of Kelvin waves. Given phase speeds of 2.8 m s$^{-1}$ for the first baroclinic mode (Gill, 1982), Kelvin waves would take about 45 days to cross the distance between the dateline and the American west coast.

In summary, positive SSH anomalies appear to be induced by buoyancy flux anomalies in the tropical Pacific during the peak of the El Niño event. Precipitation anomalies and collocated halosteric SSH anomalies suggest that fresh water fluxes play an important part in this mechanism. Off–equatorial SSH anomalies propagate westward as Rossby waves and counteract the negative SSH anomalies in the western tropical Pacific. Although, as pointed out with respect to Fig. 6, the relative contribution of halosteric and thermosteric anomalies varies over time, we find this mechanism to be relevant during most El Niño events and not only in the particular year shown here.

## 5  Summary and conclusion

We used a global ocean model to run a set of sensitivity experiments to determine the impact of ocean–atmosphere buoyancy fluxes on interannual to decadal SSH variability in the tropical Pacific. As expected, wind stress variability, associated with the major basin–scale climate modes, is the most important driver of SSH variability in the region, causing the well known zonal dipole with opposing tendencies in the western and eastern parts of the tropical Pacific. We find that SSH variability due to buoyancy fluxes is generally small but not negligible, with the strongest contributions around $10°$ N/S and a maximum in standard deviation of 2 cm in the central portion of the south Pacific.

Previous studies based on sensitivity experiments (Piecuch and Ponte, 2012; Forget and Ponte, 2015; Meyssignac et al., 2017) tend to agree with the results presented here in that they find a maximum of buoyancy flux forced interannual SSH variability on the order of 2–3 cm in the central tropical Pacific south of the equator. Studies using different methods, including budget analyses of ocean state estimates, identified an impact of buoyancy flux forcing mostly in the western tropical Pacific

(Piecuch and Ponte, 2011; Fukumori and Wang, 2013), with amplitudes of the related sea level variability of up to 10 cm. However, all previous studies were limited to a period of 2-3 decades, i.e., the altimetric period since 1992.

By building on model simulations over the last six decades, the present study provides an extended perspective on the variability on interannual time scales and also allows a view on the decadal–scale changes. While previous studies mostly focus on heat fluxes, we also analysed the importance of fresh water fluxes for steric sea level anomalies.

The results of the sensitivity experiments presented here suggest that, in contrast to the prevailing wind stress driven zonal dipole, buoyancy flux driven sea level variability is zonally uniform across the extent of the basin but is correlated with the wind stress driven part. More specifically, buoyancy fluxes tend to dampen SSH variability in the western part of the basin but amplify it in the east. The eastern part of the domain is mostly dominated by interannual heat flux variability that increases during El Niño events, thereby increasing the thermosteric SSH and amplifying positive sea level anomalies in the region. Here, buoyancy forcing has no impact on the variability on decadal timescales. Halosteric sea level variability driven by buoyancy fluxes is negligible in this area, since its variability is largely confined to the central and western parts of the domain. In contrast, steric SSH variability due to surface buoyancy fluxes in the central and western Pacific is equally affected by thermosteric and halosteric contributions, with an increasing, relative importance of halosteric SSH in the last three decades due to a reduction of thermosteric SSH variability by more than 20%. Overall, buoyancy fluxes contribute to SSH variability on both interannual and decadal timescales, with a higher relative importance for the latter where they dampen the wind stress driven signal by 20%.

A source of uncertainty for the analysis presented here is intrinsic variability. We found it to be negligible in the tropical Pacific where it accounts for less than 5% of the interannual variability. This is in line with previous studies that showed that intrinsic variability accounts only for a small fraction, mostly well below 10%, of the interannual (Penduff et al., 2011; Sérazin et al., 2015; Close et al., 2020; Carret et al., 2021) to decadal (Sérazin et al., 2016; Llovel et al., 2018) SSH variability in the tropical Pacific.

Yet, another source of uncertainty is a possible non–linear response of the ocean to the atmospheric forcing. Indeed, based on a state–estimate analysis, (Piecuch and Ponte, 2012) showed that such effects might be relevant in the tropical Pacific, at least on a local scale. Although, with the methodology used here, it is difficult to quantify such effects, we found indications for non–linearities in particular in the western tropical Pacific.

The example of the 1997/98 El Niño event illustrates the oceanic response to buoyancy forcing. Halosteric SSH anomalies are induced in the central tropical Pacific by strong precipitation anomalies during El Niño events and propagate westward as off–equatorial Rossby waves where they act to dampen negative, thermosteric SSH anomalies. The oceanic response to buoyancy forcing can therefore be regarded as a dynamical signal that allows local buoyancy flux anomalies to remotely impact SSH. The mechanism appears similar to that described by Piecuch and Ponte (2012), suggesting a robust result of

model sensitivity experiments to probe the relative importance of different forcing components. The contribution of both heat

and freshwater fluxes identified here also emphasizes the need for an accurate representation of all components of the local

air-sea buoyancy fluxes in model studies of regional sea level projections in the tropical Pacific.

*Data availability.* Data shown in this paper are available at https://data.geomar.de/downloads/20.500.12085/2832842a-6bd6-4f38-b4f6-741a5a65399d/

*Author contributions.* PW and CB defined the overall research problem and methodology; PW and MS developed, ran, and validated the OGCM experiments ; PW produced all figures; PW prepared the paper with contributions from all coauthors.

*Competing interests.* The authors declare that they have no conflict of interest.

*Acknowledgements.* PW was supported by the Deutsche Forschungsgemeinschaft (DFG) as part of the Special Priority Program (SPP)–1889 "Regional Sea Level Change and Society" (Grant BO907/5-1). All model simulations have been performed at the North–German Supercomputing Alliance (HLRN). We thank two anonymous reviewers for their constructive comments.

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

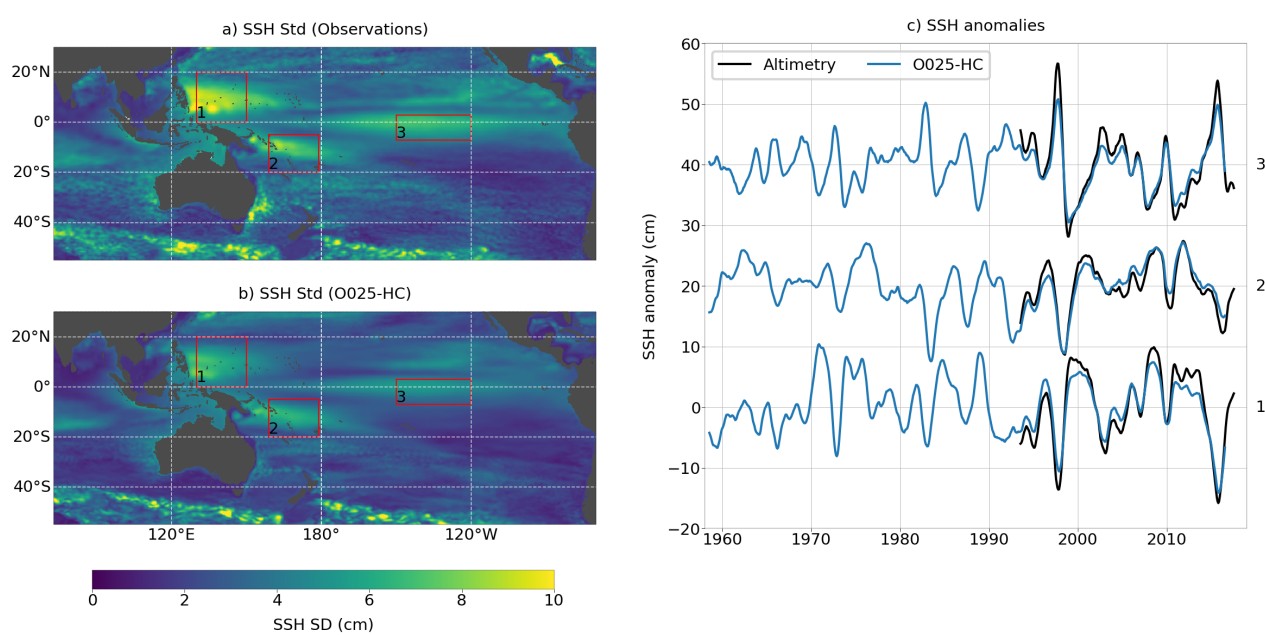

**Figure 1.** Standard deviation (SD) of SSH from (a) observations and (b) model simulation O025-HC. (c) SSH anomalies from observations and O025-HC averaged over three areas indicated by red boxes on the maps. Note that in panel a constant offset of 20 cm and 40 cm was added to anomalies from box 2 and 3 respectively. SD is based on period from 1993 to 2016. The global mean trends have been subtracted, and all data was smoothed with a 12–month running mean window prior to calculation.

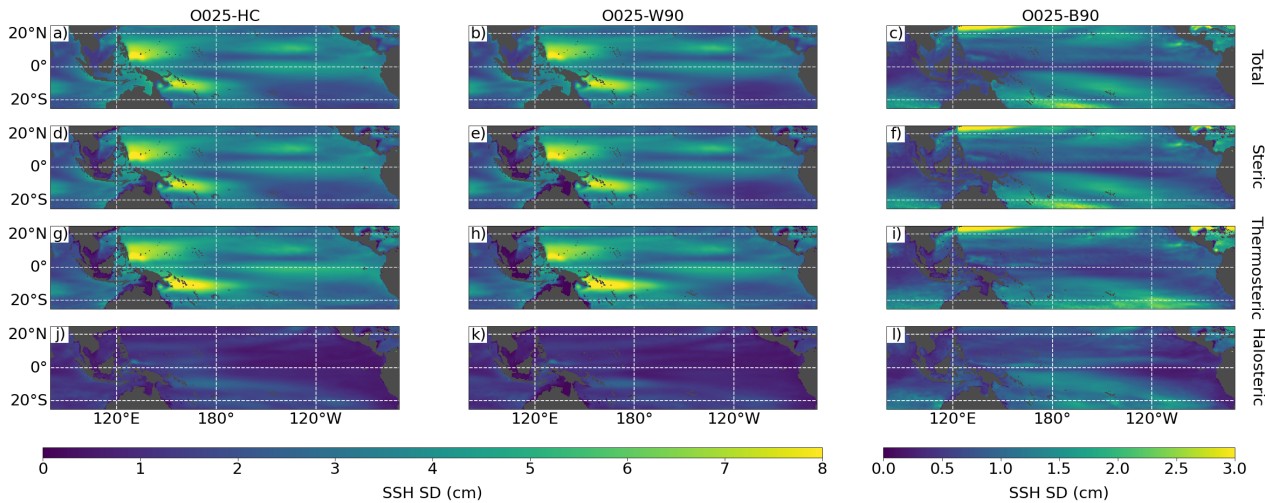

**Figure 2.** SD of total SSH (first row), steric SSH (second row), thermosteric SSH (third row) and halosteric SSH (fourth row) from O025-HC (left), Wind90 (middle) and Buoy90 (right). SD is based on the period from 1958 to 2016. The global mean trends have been subtracted, and all data was smoothed with a 12–month running mean window prior to calculation.

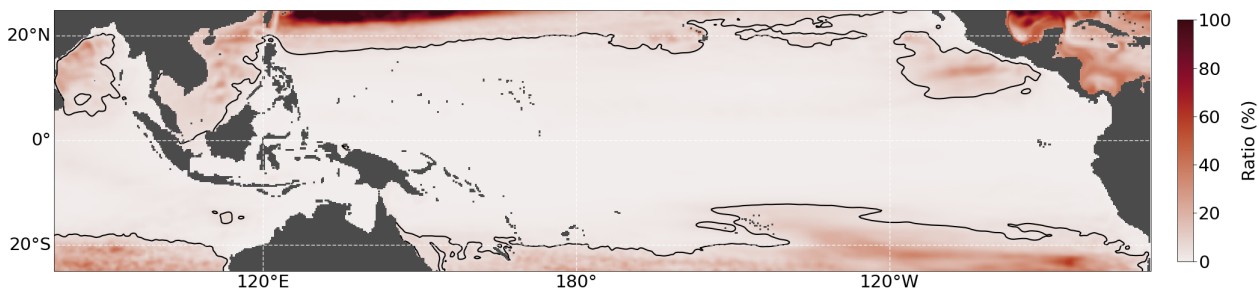

**Figure 3.** Ratio of SSH variance from O025-RYF90 and O025-HC. Black contour lines marks 5% treshold. SDs are based on the period from 1958 to 2016. The global mean trends have been subtracted and all data was smoothed with a 12–month running mean window prior to calculation.

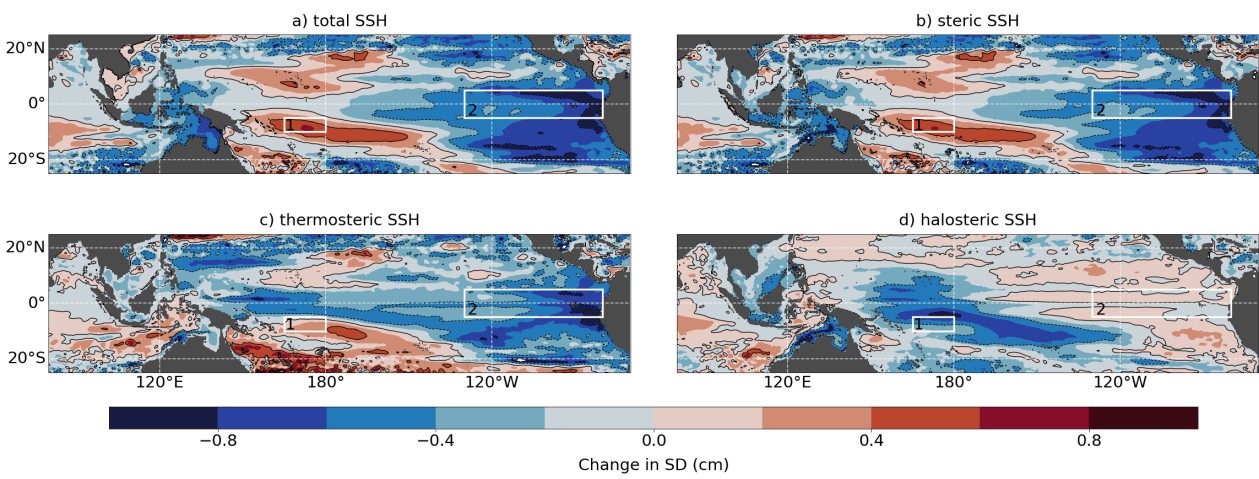

**Figure 4.** SD from O025-HC minus SD from O025-W90 for (a) total SSH, (b) steric SSH, (c) thermosteric SSH and (d) halosteric SSH. SD is based on the period from 1958 to 2016. The global mean trends have been subtracted and all data was smoothed with a 12–month running mean window prior to calculation. Contour lines mark $\pm 4$ cm, $\pm 8$ cm and 0 cm.

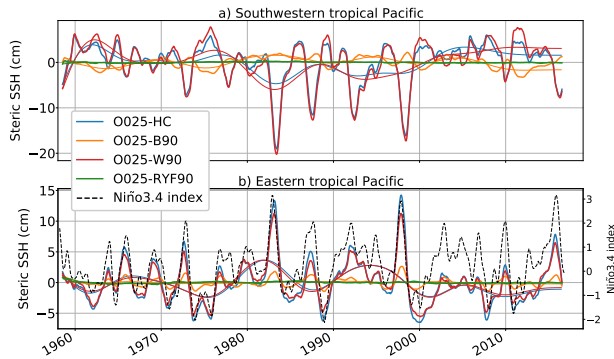

**Figure 5.** Timeseries of steric SSH anomalies from O025-HC, O025-W90 and O025-B90 averaged over boxes in the (a) southwestern and (b) eastern tropical Pacific. Exact regions are shown as red boxes (1,2) in Fig. 4. Thick lines are smoothed with a 12–month running mean. Thin lines are low–pass filtered with a 8–year butterworth filter. Dashed line in panel b) indicates Niño3.4 index.

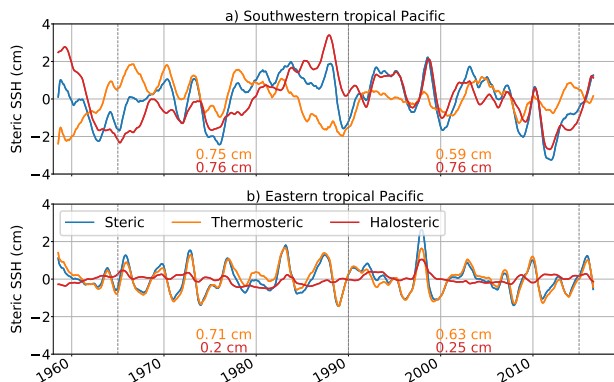

**Figure 6.** Timeseries of steric SSH anomalies from O025-B90 averaged over boxes in the (a) southwestern and (b) eastern tropical Pacific. Exact areas are shown as boxes 1 and 2 respectively in Fig. 4. Standard deviations for thermosteric and halosteric SSH anomalies are given for the periods from 1965–1990 and 1990–2015. All time series are smoothed with a 12–month running mean.

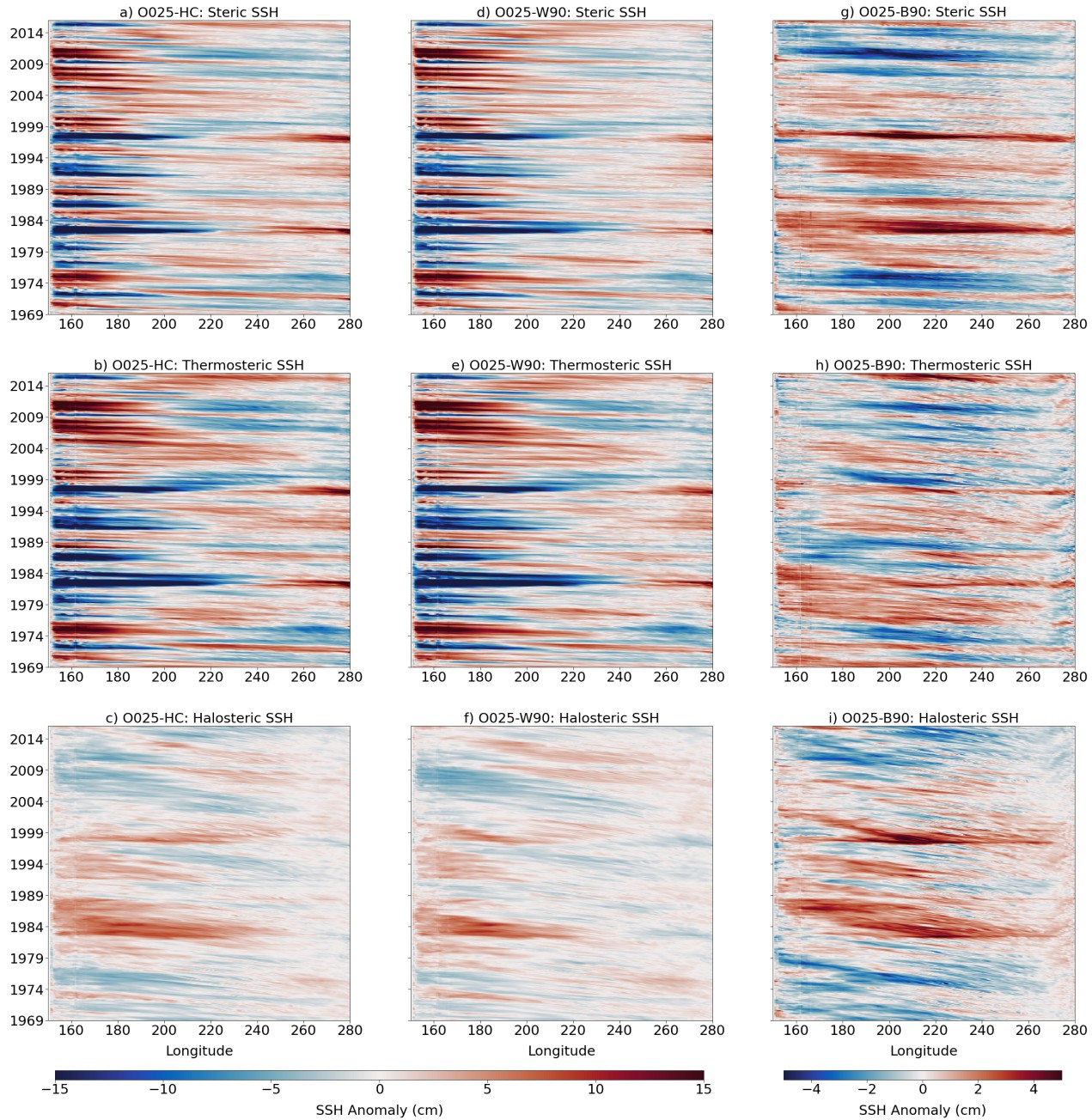

**Figure 7.** Hovmöller diagram of SSH anomalies at $10°$ S. Steric SSH (first row), thermosteric SSH (second row) and halosteric SSH (third row) from O025-HC (left), O025-W90 (middle) and O025-B90 (right). All data has been smoothed with a 12–month running mean.

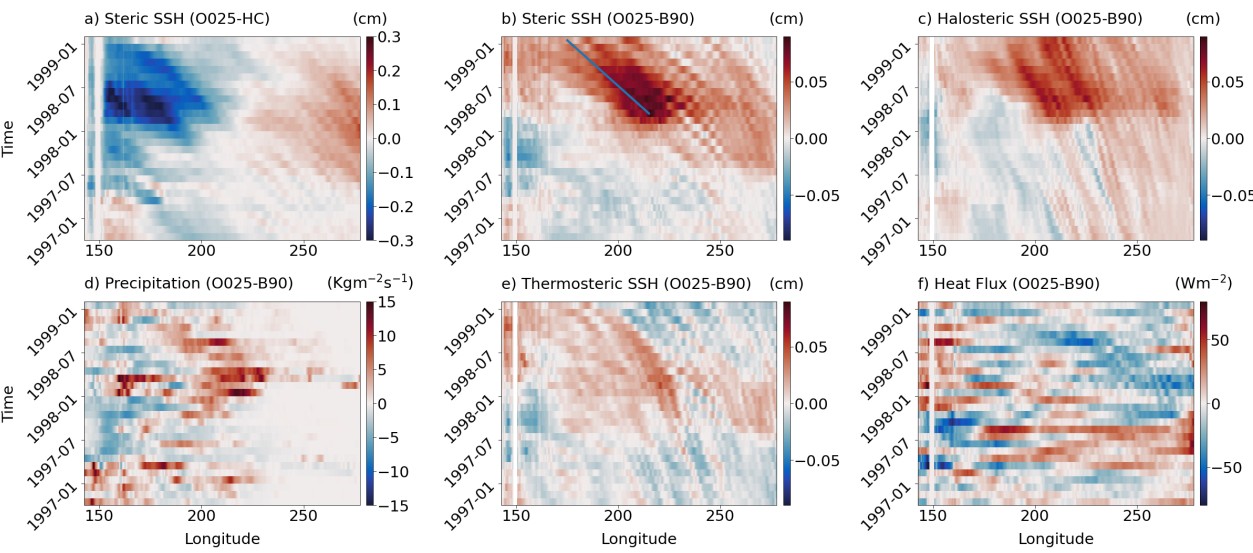

**Figure 8.** Hovmöller diagram at $10°$ S. (a) Anomalies of steric SSH from O025-HC and anomalies from O025-B90 of (b) steric SSH, (c) halosteric SSH, (d) precipitation, (e) thermosteric SSH and (f) net downward heatflux.

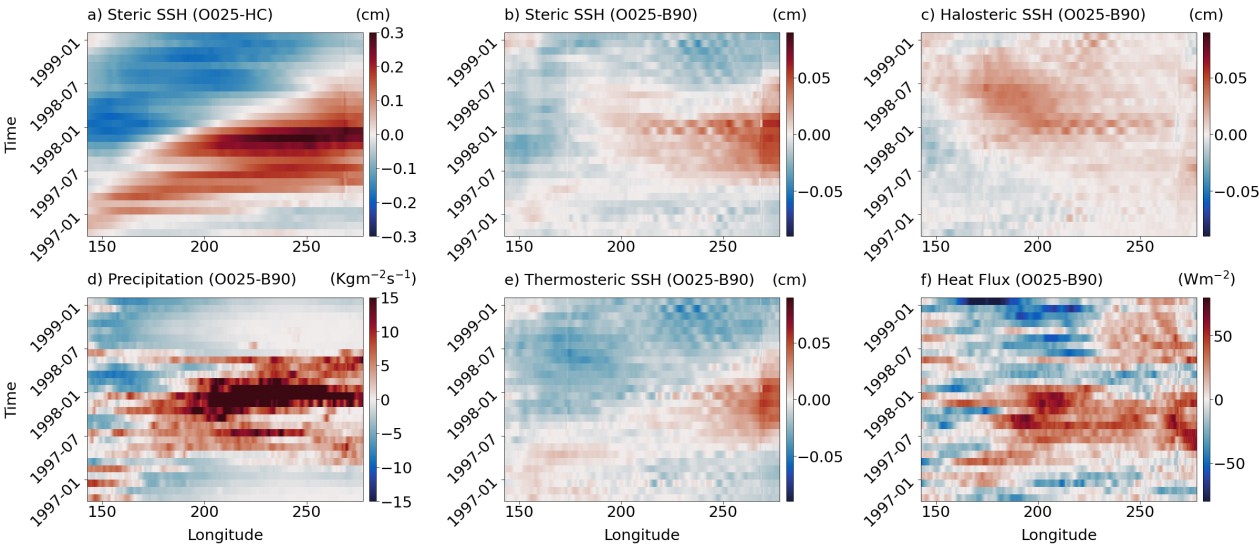

**Figure 9.** Hovmöller diagram at the equator. (a) Anomalies of steric SSH from O025-HC and anomalies from O025-B90 of (b) steric SSH , (c) halosteric SSH, (d) precipitation, (e) thermosteric SSH and (f) net downward heatflux.