# Peer review of "Contribution of buoyancy fluxes to tropical Pacific sea level variability"

_Ocean Science, 2021_

## Author Comment (AC3)

[Figure]

*Figure 3: Ratio of SSH variance from O025-RYF90 and O025-HC. Black contour lines marks 5% treshold. SDs are based on the period from 1958 to 2016. The global mean trends have been subtracted and all data was smoothed with a 12–month running mean window prior to calculation.*

---

## Author Response (AR1)

We would like to sincerely thank both referees for their time and effort they put into this review and the helpful suggestions improving our manuscript. Below we address the issues raised by the referees. Original comments by the referees are printed in bold, and specific changes to the manuscript are printed in italic.

**Referee 1**

**The experiments and results conducted by the authors to estimate the effect of buoyance fluxes on the tropical Pacific sea level were interesting and well supported the conclusions of the claims. However, it is necessary to explain some terminologies and experiments further and use a more quantified language to interpret the experimental results.**

**Major comments**

**1. There is not enough description of the buoyancy flux in the title. Readers without prior knowledge cannot understand at all. It is necessary to introduce what buoyancy flux means.**

We agree that an introducing of buoyancy fluxes is missing. We will extend the introduction accordingly.

*L35-37: However, exchange of heat and freshwater between the ocean and the atmosphere can also affect sea level variability, as they change the density of the surface waters. They are commonly referred to as "buoyancy fluxes".*

**2. The results of O025-W90 and O025-B90 represent momentum and buoyancy flux effects, respectively. However, there is too little explanation for each of these cases. Authors need to explain to both models so that readers can get a rough understanding of what each of these results means without reading the bibliography.**

Point taken. In addition to a proper introduction of buoyancy fluxes we will extend the model description of the sensitivity experiments and clarify the purpose of these experiments, add specifics on the method and point out limitations of this sort of experiments.

*L75-94: This requires a method to eliminate interannual variability from the forcing, and we followed the approach of a "repeated year forcing" (Stewart et al. 2020) to construct quasi--climatological atmospheric fields. Specifically, a 12--month period, that is "neutral" with respect to a range of climate indices (hence the term quasi--climatological), is taken from the JRA55-do dataset and used repeatedly for the computation of turbulent fluxes. We followed the recommendation from Stewart et al. 2020 and used the period from May 1990 to April 1991 to force 59 cycles of each sensitivity experiment to match the length of the hindcast. The transition is moved from April to May, rather than December to January, to avoid periods of high variability at high latitudes, and therefore abrupt changes. The reader is referred to Stewart et al. 2020 for further information regarding the repeated year approach. All model output is stored with monthly resolution and has been used previously for studies of Indian Ocean heat content (Ummenhofer et al., 2020) and marine heatwaves (Ryan et al., 2021).*

*Three caveats need to be considered when analysing the sensitivity experiments: firstly, the computation of turbulent fluxes does not only depend on the atmospheric forcing but also the state of the underlying ocean. This introduces a source of interannual variability of buoyancy and momentum fluxes even where we aim to suppress it. Secondly, all experiments contain intrinsic variability, which is generated spontaneously by the ocean rather than by the atmospheric forcing.*

*OGCMs with an eddy--permitting resolution, as the ones used here, are able to capture this variability (Penduff et al. 2011). Consequently, variability in O025-W90 for example, can not exclusively be attributed to momentum flux but might also be generated by intrinsic variability. We will use the climatological experiment O025-RYF90, where this will be the dominant source of variability, to quantify this. Thirdly, the approach assumes a linear superposition of variability forced by momentum and buoyancy fluxes. However, the OGCM used here does also include nonlinear responses of the ocean to atmospheric forcing, for example due to the non--linear nature of the equation of state, which may violate this assumption.*

**3. There are more strong ENSO events 1982/83, 2015/16. If the authors provide one more analysis, the authors will be able to convey the claims more clearly. There is a limit to reaching generalized conclusions through a single case study.**

We agree that it is useful to check if the mechanism identified for the 1997/1998 El Nino are also active during other events. In general, we find this to be true as buoyancy flux forced anomalies appear during most ENSO event in the central, tropical Pacific. The relative contributions of halosteric and thermosteric anomalies varies over time, but this is also visible in figure 6 and has been discussed there. We failed to mention this in the manuscript and will extend it accordingly.

*L238-240: Although, as pointed out with respect to Fig. 6, the relative contribution of halosteric and thermosteric anomalies varies over time, we find this mechanism to be relevant during most El Nino events and not only in the particular year shown here.*

**4. Figures 6, 7, and 8 are very important. However, it is confusing because the pictures are not properly marked. Add lines and boxes to make the pictures easier to read.**

We will add additional labels to make them easier to read.

*Please see the new figures 7,8,9 (old figures 6,7,8) in the revised manuscript.*

**5. I generally understand and agree with the authors' claims. By the way, the language used for comparison is not clear (especially section 3 results). It is necessary to quantify the comparison, and if it is difficult to quantify, please provide more details in what respects they are similar or dissimilar, or triggered.**

We see the point and suggest three modifications in this respect:

1. Quantify the change of variability shown in figure 3 and give numbers in the text. Also modify figure 3 to show a discrete colormap that enables the reader to infer values from the figure and follow our argument. This also relates to your comment below.

2. Quantify the impact of buoyancy fluxes by giving the relative change in variability between O025-HC and O025-W90. We already included this for the low-frequency variability in the western box and will extend it to both boxes and also to the interannual variability.

3. We already compared the amplitudes of variability for halosteric and thermosteric SSH shown in figure 5. We suggest extending this comparison by giving correlation coefficients.

*L145-149 A horseshoe shaped minimum is located at the eastern boundary, where variability is reduced by up to 1 cm in the very east. Positive values are centred at 175°E on both sides of the equator at 10° latitude with values of up to 0.4 cm in the north and 0.6 cm in the south. The same pattern can be seen for steric SSH (Fig. 4b) and, to a slightly lesser extent, for thermosteric SSH*

*(Fig.4c). The change in halosteric SSH is similar in magnitude but shows a different pattern. Changes are mainly limited to western and southwestern tropical Pacific, where the halosteric SSH variability decreases in O025-W90 by up to 0.8 cm.*

*L159-160: Measured by the change in SD between O025-HC and O025-W90, buoyancy fluxes dampen the variability by 11% on interannual and by 20% on decadal timescales. (Fig. 5a).*

*L166-167: This is in line with the amplifying effect of buoyancy fluxes described above, and we find an increase of variability due to buoyancy fluxes by 16%.*

*L176-177: Correlation coefficients are 0.62 and 0.48 (both significant on a 99% confidence interval)*

*L181-182: Steric SSH in the eastern equatorial Pacific is almost exclusively governed by thermosteric SSH with correlation coefficients of 0.89 (significant on a 99% confidence interval)*

**Miner comments**

**L21: Add a description of "Ocean atmosphere buoyancy fluxes." before using this term.**

We replaced the term buoyancy fluxes at this early stage by heat and freshwater fluxes and introduced it properly in the second to last paragraph of the introduction.

**L24: Define SLC before using.**

Following a suggestion by referee #2 we avoided the term SLC altogether as it is commonly used for decadal to multidecadal variability and trends. We used the term "sea level variability" in this case.

**L76: Is the meridional dipole right what the authors are trying to explain? It seems to be explaining the zonal dipole. If the authors try to explain the meridional dipole, please make it more clear.**

Thanks for catching this error. It should read "zonal dipole".

**L84: "In all cases, the correlation coefficient is over 0.95", but in the case of "3", 0.95 is unreasonable. Please check it.**

We double checked the correlation and can confirm that correlation for box "3" is at 0.98. However, the correlation for box 2 is only 0.93. We will correct the sentence accordingly.

**L91: it is helpful to show SPCZ on the map.**

We would like to keep the figures as simple as possible and avoid adding additional lines. Instead, we suggest to remove the reference to the SPCZ in the text and refer to the region as "southwestern tropical Pacific". This would also be consistent with the preceding paragraph and the naming of the boxes in figure 1.

**L97-98: A bibliography is needed.**

Yes, we will include references.

**L115-116: "Changes are mainly limited… in O025-W90." It is difficult to agree with the argument by judging by the colors only.**

We changed the colormap to a discrete colormap and marked individual steps with contour lines.

*Please see the revised figure 4 (former figure 3) in the manuscript.*

**L133-137: It is difficult to accept the argument from a comparison of two temporal windows only. It is recommended to make a moving calculation window and show the change of SD.**

A moving windows analysis gives a decline of SD of thermosteric SSH of about 0.1cm/10years since around 1985. We will include this in the text. Because this is only a minor finding which we do not pursue any further in this study, we suggest to not add another plot.

*L179-180: (SD drops from 0.75 cm to 0.59 cm and a moving window calculation yields a linear decline of SD 0.1 cm per 10 years that starts around 1985; Not shown)*

**L139-141: Please provide a visual comparison with the ENSO index (simply just add up any ENSO relating indices). There is a limit to generalizing to only one event.**

We agree that this would be helpful. We will add the Nino34-index computed from the model to figure 6b)and point out the correlation in the text.

**L167-168: "These anomalies… . (Fig. 7b)." I don't understand. An additional explanation is required.**

Sorry for the poor phrasing. We will reformulate this sentence to be more concise.

*L212-213: Once they reach the western part of the basin by mid–1998, these positive anomalies tend to reduce the prevailing negative anomalies (Compare fig. 8a and b)*

**Figure 1: "SD" needs to be predefined before use.**

Corrected

**Referee 2**

**The authors revisit the effects of surface buoyancy fluxes on sea level variability in the tropical Pacific, a topic that has received comparatively little attention in the literature. Following up on Piecuch and Ponte (2012) and others, they use three different numerical experiments with an eddy-permitting (1/4 degree horizontal grid) to separate out the role of surface wind and buoyancy forcing over an extended period (1958-2016). The results are an interesting contribution to the literature, confirming the importance of buoyancy fluxes in several areas of the tropical Pacific, their excitation of Rossby waves and related dynamic sea level signals, and pointing out the influence of both heat and freshwater fluxes at different (interannual to decadal) time scales.**

**The conclusions of the paper are reasonably well supported by the analyses shown, but there are a couple of issues that need to be discussed in the manuscript. In trying to separate wind**

**and buoyancy effects using the experiments with full and climatological forcing described in section 2, there is always the issue of nonlinearity, as discussed for example by Piecuch and Ponte (2012). Moreover, given the eddy-permitting nature of the runs used, it is also not clear how much of the differences in variability between runs with full and climatological forcing are due to "chaotic intrinsic" eddy-related processes as discussed by Carret et al. (2021) and references therein (not cited in the current manuscript).**

**Piecuch and Ponte (2012) seem to imply significant nonlinear effects in some of the regions discussed in the current paper. Carret et al. (2021) point to generally weak effects of intrinsic variability relative to atmospherically forced variability in the tropical Pacific at interannual time scales. Although the three runs used by the authors do not permit addressing these isues, the manuscript should nevertheless acknowledge and discuss them explicitly.**

**[Carret, A., Llovel, W., Penduff, T., & Molines, J.-M. (2021). Atmospherically forced and chaotic interannual variability of regional sea level and its components over 1993–2015. Journal of Geophysical Research: Oceans, ⬛126, e2020JC017123. https://doi. org/10.1029/2020JC017123]⬛**

We will include the results of a climatological experiment (O025-RYF90), where we used the repeated year forcing approach to replace the entire atmospheric forcing, to address this issue. This experiment allows us to quantify the role of intrinsic variability. We found that the intrinsic variability accounts for less than 5% of the interannual variability in the tropical Pacific and we will include an additional figure to show this.

Quantifying non-linear effects is not straight forward. In the absence of non-linear effects and intrinsic variability, the linear superposition of anomalies from O025-B90 and O025-W90 should be equivalent to the results from O025-HC. We can get a rough estimate of the importance of non-linear effects by comparing the root-mean-square error ( RMSE(HC-W90-B90) ) to the SD of the climatological experiment. We did this for the anomalies shown in Fig. 3, and we found that the RMSE outweighs the SD of RYF90 in the western tropical Pacific but not in the eastern basin. This suggests that non-linear effects might be important in the western part of the basin. We will include this in the manuscript and also discuss our findings with respect to existing literature.

*Additional figure 3 in the revised manuscript.*

*L85-94: Three caveats need to be considered when analysing the sensitivity experiments: firstly, the computation of turbulent fluxes does not only depend on the atmospheric forcing but also the state of the underlying ocean. This introduces a source of interannual variability of buoyancy and momentum fluxes even where we aim to suppress it. Secondly, all experiments contain intrinsic variability, which is generated spontaneously by the ocean rather than by the atmospheric forcing. OGCMs with an eddy--permitting resolution, as the ones used here, are able to capture this variability (Penduff et al. 2011). Consequently, variability in O025-W90 for example, can not exclusively be attributed to momentum flux but might also be generated by intrinsic variability. We will use the climatological experiment O025-RYF90, where this will be the dominant source of variability, to quantify this. Thirdly, the approach assumes a linear superposition of variability forced by momentum and buoyancy fluxes. However, the OGCM used here does also include nonlinear responses of the ocean to atmospheric forcing, for example due to the non--linear nature of the equation of state, which may violate this assumption.*

*L132-136: Note that O025-B90 does also show strong variability in the boundary current region of the Kuroshio, north of 20°N. This is a region of strong intrinsic variability (Fig. 3), most likely due to the strong mesoscale activity. We will therefore neglect this region in our analysis. It is also evident from Figure 3 that intrinsic variability is mostly negligible equatorward of approximately 20° latitude. Here, the ration between SSH variance from O025-RYF90 and O025-HC does not exceed 5%.*

*L160-164: The superposition of anomalies from O025-W90 and O025-B90 should reconstruct anomalies from O025-HC. We find a root–mean–square–error(RMSE) of 0.45 cm for this reconstruction. As discussed in section 2, possible causes are intrinsic variability (green lines in Fig. 5) and non-linear effects. Intrinsic variability is weak (SD of 0.11 cm), which suggest that non-linearities are the main cause.*

*L168-170: Here, the superposition of O025-B90 andO025-W90 to reconstruct O025-HC gives a much smaller RMSE of 0.2 cm. Unlike in the western box, intrinsic variability can account for a large fraction of this (SD of 0.14 cm).*

*L273-282: A source of uncertainty for the analysis presented here is intrinsic variability. We found it to be negligible in the tropical Pacific where it accounts for less than 5% of the interannual variability. This is in line with previous studies that showed that intrinsic variability accounts only for a small fraction, mostly well below 10%, of the interannual (Penduff et al., 2011; Sérazin et al., 2015; Close et al., 2020; Carret et al., 2021) to decadal (Sérazin et al., 2016; Llovel et al., 2018) SSH variability in the tropical Pacific.*

*Yet, another source of uncertainty is a possible non-linear response of the ocean to the atmospheric forcing. Indeed, based on a state–estimate analysis, (Piecuch and Ponte, 2012) showed that such effects might be relevant in the tropical Pacific, at least on a local scale. Although, with the methodology used here, it is difficult to quantify such effects, we found indications for non-linearities in particular in the western tropical Pacific.*

**The manuscript contains many typos and careless errors, repeated at places several times. I have tried to point these out in the long list below, although I probably did not get them all. Needless to say, the authors should have proofread their manuscript more carefully and need to do that before submitting a revised version.**

We apologize and will carefully proofread our manuscript before resubmission.

**Other comments by line number**

**l11 Delete comma after "both"**

Corrected

**l16-17 Broken sentence: I suggest a colon, instead of a period after "processes". In addition, "melting of land ice" is not the only reason the ocean's total mass changes. Imbalances in precipitation, evaporation and river runoff can also be contributors, depending on time scale.**

We will modify the sentence accordingly.

**l24 Sea level change (SLC) normally refers to long term (multidecadal or centennial) variability. Here and elsewhere in the paper, perhaps you want to use the more general term of sea level variability, which can include shorter time scales of relevance to the paper.**

We will change this throughout the manuscript.

**l29 Define all acronyms on first mention.**

Corrected

**l48 "…that allow…"**

Corrected

**l53 "…ocean general circulation…"**

Corrected

**l63 "…a relaxation timescale…" and remove comma after "correction"**

Corrected

**l70 "May 1990"**

Corrected

**l65-71 Not exactly clear what the forcing is and why May 1990 to April 1991 is chosen. In particular, forcing could still contain interannual variability (e.g., if there is a long term trend, it will have a jump at the wrapping date of April 30, which adds energy at most frequencies including interannual). I guess the particularly period chosen is trying to avoid these effects, but there should be more explicit discussion of these issues in the paper.**

The period from May 1990 to April 1991 (rather than January to December) is indeed chosen to minimize sudden changes in the forcing that might introduce spurious transients. The year is chosen because it resembles a "neutral" year with respect to several climate indices such as SOI, NAO and SAM. Stewart et al. 2020 tested three different periods (1984, 1990, 2003) with three different model configurations. They find that inter-model differences are much larger than inter-forcing differences between these three periods. They conclude that the choice is therefore not critical but give a general recommendation for the 1990-1991 period. We followed this recommendation.

We would like to avoid a detailed description of the procedure and its motivation but refer to Stewart et al. 2020 instead. However, we agree that the paragraph is difficult to understand for someone not familiar with the cited references. We will rephrase it accordingly and give more details.

*L75-83: This requires a method to eliminate interannual variability from the forcing, and we followed the approach of a "repeated year forcing" (Stewart et al. 2020) to construct quasi--climatological atmospheric fields. Specifically, a 12--month period, that is "neutral" with respect to a range of climate indices (hence the term quasi--climatological), is taken from the JRA55-do dataset and used repeatedly for the computation of turbulent fluxes. We followed the recommendation from Stewart et al. 2020 and used the period from May 1990 to April 1991 to force 59 cycles of each sensitivity experiment to match the length of the hindcast. The transition is moved from April to May, rather than December to January, to avoid periods of high variability at high latitudes, and therefore abrupt changes. The reader is referred to Stewart et al. 2020 for further information regarding the repeated year approach. All model output is stored with monthly resolution and has been used previously for studies of Indian Ocean heat content (Ummenhofer et al., 2020) and marine heatwaves (Ryan et al., 2021).*

**l74 Capital B on Boussinesq**

corrected

**Figure 1 The reader needs to be told what altimeter data is used (the link to CMEMS is not enough), and whether the model results in (b) are calculated over the same altimeter period. It is also awkward to say "interannual SD of SSH". What you have is SD of SSH series that have been smoothed with 12-month running mean.**

We will include a complete reference to the altimetry data and provide additional information in the caption of figure 1.

**l76 "Meridional dipole" should be "zonal dipole" as used in the rest of the paper.**

Corrected

**l81 Any criterion for choosing these particular boxes, other than being generally over the regions of enhanced variability? Are the results sensitive to box boundaries? This could be discussed in the text.**

We chose these boxes as we consider them representative for the variability in the region, but we acknowledge that the choice is somewhat arbitrary. However, the result is not sensitive to the exact choice of the boxes, as long as they do not cover the boundary current regions with high mesoscale activity. There, the model performance is reduced due to an insufficient spatial resolution. We will include this in the text.

*L108-111: Within a latitudinal band of about 40° around the equator, this result is not overly sensitive to the choice of the boxes (not shown). However, likely due to an insufficient spatial resolution, the model does not capture the mesoscale activity of the western boundary current regions properly and therefore underestimates SSH variability in this region even further, as can be seen by comparing Fig. 1 a) and b).*

**l99 Cite relevant works.**

Corrected

**l100-105 What about the maximum values seen in the northern most latitudes of the domain shown in fig 2?**

We neglected this region in our analysis because of intrinsic variability. The figure above illustrates this. We will point this out in the manuscript.

*L133-136: This is a region of strong intrinsic variability (Fig. 3), most likely due to the strong mesoscale activity. We will therefore neglect this region in our analysis. It is also evident from Figure 3 that intrinsic variability is mostly negligible equatorward of approximately 20° latitude. Here, the ration between SSH variance from O025-RYF90 and O025-HC does not exceed 5%.*

**l108 "assess"**

Corrected

**figures 1,2,3 The color of land is rather similar to actual values being plotted. The land could use some other less confusing color. I assume all the plots are based on 12-month smoothed series as in fig 1, but this should be made clear in the text or captions.**

Changed accordingly

**l109 Why "absolute change"? Not clear what is meant by "absolute".**

Absolute changes in contrast to relative/percental change. However, we agree that the attribute is not needed and might cause confusion. Will be removed.

**figure 3 Caption should state SD of x minus SD of y. This way the reader can be clear on what the sign of the values means.**

Changed accordingly

**l110 Move "is removed" after "forcing" on l111.**

**Corrected**

**l113 "…E on both sides…"**

Corrected

**l119 "…effect of halosteric and thermosteric SSH…"**

Changed accordingly

**l124 "In phase" means correlation. If they are anticorrelated, you should use out-of-phase.**

Corrected

**l127-128 In this case, the phase/correlation statement is redundant and should be rephrased.**

Changed accordingly

**figure 4 caption Not 12-year but 12-month running mean.**

Corrected

**l130 I would say figure 2 only suggests this interpretation. You have not done the experiments to strictly separate the effects of heat and freshwater fluxes and determine if both play a role.**

We will change "indicates" to "suggests".

**l136 Should be "0.75 cm and 0.59 cm"**

Indeed

**figure 5 caption Again you mean "12-month" running mean?**

Corrected

**l139-140 Again correct the numbers.**

Corrected

**l162 Somewhat odd numbering of one section 3 with only one subsection 3.1. I would number this section 4.**

Changed

**l156-158 Refer to the relevant figures behind this summary statement for the benefit of the reader.**

Changed accordingly

**l164 Text should clarify how the "anomalies" are defined.**

Anomalies are deviations from the seasonal climatology. We will include the definition.

**l168-171 There is an implicit assumption here that freshwater flux is the only way to generate halosteric anomalies, but that is not necessarily true. For example, heat flux could drive flows that advect both temperature and salinity fields and generate salinity anomalies. In fact, the observed compensation between halosteric and thermosteric anomalies suggests some adiabatic advective mechanism along isopycnals.**

This is of course true and we will clarify this.

*L220-222: The co-occurrence in this example suggests that the halosteric SSH anomalies are at least partly driven by fresh water fluxes. However, buoyancy fluxes in general could also change the flow field and thereby cause adiabatic temperature and salinity anomalies, that would also result in corresponding SSH anomalies.*

**l176 Refer to Fig. 8b, not 8c?**

Corrected

**l182-183 This seems to be the first mention of monthly output used for the analyses. The information should be provided much earlier in the paper (section 2).**

We will include this in the model description.

L82: *All model output is stored with monthly resolution […].*

**l210 "varies in phase"…see comments on l124.**

Corrected